# OPTIMIZING ACTIVATIONS BEYOND ENTROPY MINIMIZATION FOR TEST-TIME ADAPTATION OF GRAPH NEURAL NETWORKS

## ABSTRACT

Test-time adaptation (TTA) of classification models, which aims to optimize classifiers without labeled training samples, often employs entropy minimization as a key objective. While this approach addresses the relationship between model performance and prediction confidence or cluster structure, it can lead to model collapse due to the lack of ground truth labels. This work optimizes activations within batch normalization (BN) layers for TTA of graph neural networks (GNNs). Our proposed method optimizes BN activations in a two-step process. First, we determine weights and masks for the empirical batch mean and variance, considering training and test data statistics. Subsequently, we refine the scale and shift parameters of the BN layers using a reformulated loss function incorporating an energy-based model, aiming to enhance the model's generalization capabilities. Our approach leverages pseudo-labels derived from test samples to mitigate the potential forgetting of training data. Empirical evaluation across seven challenging datasets demonstrates the superior performance of our approach compared to state-of-the-art TTA methods.

## 1 INTRODUCTION

We study test-time adaptation (TTA) on graph neural networks (GNN) in this paper. TTA addresses the issue of the model's performance degrading when deployed in a scenario where the target (test) data differs from the training data. This discrepancy restricting the model's generalization lies in the data distribution shift between the training and the test data. TTA handles the issue by adjusting or fine-tuning the model with respect to the characteristics of the test data during inference before making predictions.

Most TTA methods are designed under the context of image classification, where the data distribution shift is usually a consequence of natural variation (Koh et al., 2021) or corruption (Hendrycks and Dietterich, 2019). Under the assumption that the unknown distribution shift is caused by the combination of known variates or corruptions across all domains (Gao et al., 2023), these TTA methods do not work well on graph data where the distribution shift is complicated. Various data distribution shifts exist (Quiñonero-Candela et al., 2022) when deploying GNN models in an environment inconsistent with the one during training, such as full-distribution shift, covariate shift, class-prior shift, class-conditional shift, or simply noise.

One SOTA method that could avoid shift-type identification is entropy minimization (EM) (Wang et al., 2021; Press et al., 2024). EM-based methods adapt classifiers by iteratively updating the model's parameters to minimize the entropy of the model's predictions, i.e., to maximize the likelihood of the observed data belonging to the most likely classes. However, entropy minimization-based methods have limitations. They often fight hard against the catastrophic forgetting of the ground truth in training data. Due to the lack of ground truth labels, they can further introduce error signals, leading to increased sensitivity to the learning rate and potential issues like model collapse.

The general guideline is to maintain a small divergence between the pre- and post-adaptation models, thereby retaining the model's inference capabilities acquired from the training data. Common approaches to implementing the general guideline include: (1) fine-tuning only a subset of parameters, such as those within batch normalization (BN) layers, (2) introducing regularization based on the

distance between pre- and post-adaptation parameter values, (3) limiting training epochs or employing small learning rates during adaptation.

In this paper, our approach for GNN is somewhat similar to optimizing activations in batch normalization layers. A BN layer comprises four parameters: mean ($\mu$), variance ($\sigma^2$), scale ($\gamma$), and shift ($\beta$). The statistics $\mu$ and $\sigma^2$ are derived from activations within a batch and maintained by moving averages for normalization purposes. The parameters $\gamma$ and $\beta$ are learned to optimally scale and shift activations, thereby enhancing the model's expressive power. Many TTA methods focus on adjusting only the statistics $\mu$ and $\sigma^2$, leaving the parameters $\gamma$ and $\beta$ unchanged (You et al., 2021; Mirza et al., 2022). However, it is evident that adjusting $\mu$ and $\sigma^2$ without modifying $\gamma$ and $\beta$ may not achieve the optimal scale and shift for the updated statistics. We propose a two-step fine-tuning method to address this limitation. In the first step, we fine-tune the mean ($\mu$) and variance ($\sigma^2$) parameters based on the activation distribution to better align training and test data. Subsequently, we fine-tune the scale ($\gamma$) and shift ($\beta$) parameters based on an augmented loss function incorporating an energy-based model, potentially enhancing the generalization ability of the model.

Regularizing the distance between pre- and post-adaptation parameter values may seem counterintuitive (Niu et al., 2022). This approach involves a trade-off: while we aim to modify parameters for adaptation, we also constrain the extent of these changes. This sacrifice of potential improvement is intended to preserve the ground truth from the training data, ensuring it is not entirely forgotten during adaptation. It is the same reason that most methods based on entropy minimization limit the number of training epochs during adaptation (Mounsaveng et al., 2024; Wang et al., 2021; Mummadi et al., 2021; Zhao et al., 2023) to avoid the risk of model collapse (Press et al., 2024): EM-based adaptation is effective for a few steps but eventually deteriorates performance after prolonged adaptation. They appear to save the computational cost, but determining the optimal number of adaptation steps is hard. To deal with this, we propose utilizing the predictions on test data as pseudo-labels to introduce the constraint rather than directly constraining the parameters themselves. Additionally, we employ a filtering and pruning mechanism to remove potentially incorrect and harmful pseudo-labels.

The contributions of this paper are summarized as follows:

- We introduce a data-driven, two-step TTA framework for GNNs. This approach first adapts BN layer statistics to the test data distribution. Then, it refines BN layer parameters using a joint energy-based model, overcoming the limitations of existing entropy minimization-based methods.
- We propose a data-driven method for determining optimal adaptation weights, leveraging non-parametric density estimation, the Jensen-Shannon divergence, and a learnable mask matrix to effectively balance contributions from training and test statistics. This mask matrix M allows for selective adjustment of specific dimensions within the BN layer, leading to more effective adaptation. (Section 3.1)
- We integrate an energy-based model (EBM) into our TTA framework to enhance model generalization and calibration. High-quality soft pseudo-labels are ensured through entropy-based selection and confidence-based filtering. The EBM approach contributes to more reliable adapted predictions and further enhances model calibration. (Section 3.2)

Besides, compared with popular TTA methods, the results from extensive experiments demonstrate the proposed framework's effectiveness. It is worth noting that TTA methods that do not require access to the training data are often referred to as Fully TTA methods. While our proposed method maintains a small histogram matrix to store activation distributions in BN layers, this information can be obtained solely by observing the last training epoch. Inherently, BN layers also store the activations' first and second moments (mean and variance) from the training dataset.

## 2 Test-Time Adaptation on Graph Neural Netowrks

### 2.1 Problem Statement

Let $G = \{V, E, X\}$ denote an attributed graph, where $V$ represents the set of nodes, $E$ represents the set of edges, and $X \in \mathbb{R}^{N \times d}$ is the feature matrix. Here, $N = |V|$ denotes the number of nodes, and $d$ represents the dimensionality of the features. Let $A$ be the adjacency matrix of $G$. For any two nodes $u$ and $v$, if there exists an edge connecting them, then $A_{u,v} = 1$. Otherwise, $A_{u,v} = 0$.

We evaluate test-time adaptation on the node classification task, where a distribution shift exists between the training and test datasets. Let $Y = \{y_1, y_2, ..., y_C\}$ denote the set of class labels, where $y_i$ represents one of the $C$ possible labels. $D_{tr}$ and $D_{ts}$ represent the training and test datasets, respectively. Let $\theta$ denote the parameters of the GNN-based classification model $f_\theta : G \to Y$, trained on $D_{tr}$. TTA is typically performed in an online fashion. Given a graph $G_i \in D_{te}$, TTA fine-tunes the model parameters $\theta$ before inferring the labels of nodes in $G_i$. The objective is to find improved parameters $\theta^*$ for the model $f$, such that the updated model $f_{\theta^*}$ can achieve enhanced generalization ability and superior performance on $G_i$ compared to the original model $f_\theta$.

Our approach for TTA on graph neural networks optimizes activations by fine-tuning the parameters within the batch normalization layers in the GNN architecture. We do not assume a specific type of data distribution shift, as our method operates on the activations within these layers rather than the original data space. Notably, some existing research, such as Jin et al. (Jin et al., 2022b), explores modifying the input graphs for TTA, which we will discuss further in the experimental evaluation section.

## 2.2 PRELIMINARY: BATCH NORMALIZATION

Batch normalization (Ioffe and Szegedy, 2015) is a key advancement in deep neural networks, which improves both model performance and training speed by regularizing the distribution of activations. In the training stage, let $\{x_i\}_{i=1}^b$ represent the input activations of a BN layer in a batch of size $b$. The mean ($\mu_b$) and variance ($\sigma_b^2$) of the activations are calculated as follows: $\mu_b = \frac{1}{b}\sum_{i=1}^b x_i$, and $\sigma_b^2 = \frac{1}{b}\sum_{i=1}^b (x_i - \mu_b)^2$. The BN operation is then performed as shown in Eq. 1, where $\gamma$ and $\beta$ are learnable scale and shift parameters used to optimize the distribution of activations. $\varepsilon$ is a small constant added to the denominator to prevent division by zero.

$$x'_i = \frac{x_i - \mu_b}{\sqrt{\sigma_b^2 + \varepsilon}} \cdot \gamma + \beta. \tag{1}$$

During training, the parameters ($\hat{\mu}, \hat{\sigma}^2$) in BN layers are maintained using a moving average to capture the overall statistical information of the training samples as follows: $\hat{\mu}_k = (1 - \rho) \cdot \hat{\mu}_{k-1} + \rho \cdot \mu_b$, and $\hat{\sigma}_k^2 = (1 - \rho) \cdot \hat{\sigma}_{k-1}^2 + \rho \cdot \sigma_b^2$, where $\hat{\mu}_0 = 0$, $\hat{\sigma}_0^2 = 1$. The momentum parameter $\rho$ controls the update rate of these values. At test time, These maintained values, which remain fixed at test time, are then utilized to normalize the activations during inference, with the same equation as in Eq. 1.

BN layers are crucial in modern GNNs, contributing to improved model training stability. Many state-of-the-art GNN architectures incorporate BN layers (Xu et al., 2019; Jin et al., 2022b; Wu et al., 2022). In GNNs, BN is usually applied after each GNN layer, with the input to the BN layers being the $i$-th GNN layer embeddings $H^{(i)}$. The normalized representations effectively stabilize the output of each GNN layer and avoid overflow of popular aggregation functions in deep GNNs (Li et al., 2019).

## 3 PROPOSED METHOD

Recall that BN layers comprise four parameters: mean ($\mu$), variance ($\sigma^2$), scale ($\gamma$), and shift ($\beta$). We can divide these parameters into two groups:

- Statistic group: The mean ($\mu$) and variance ($\sigma^2$) are estimated from the data and capture the statistical properties of the activations within a batch.

- Parameter group: The scale ($\gamma$) and shift ($\beta$) are learnable and optimized by the loss. They allow the model to adjust the normalized activations to better suit the task at hand.

Our approach for TTA on GNNs optimizes activations within BN layers by fine-tuning the statistic and parameter groups separately. The process comprises two successive steps: batch normalization statistic adaptation (BNSA) and batch normalization parameter adaptation (BNPA). The overall process of BNSA and BNPA is presented in Fig. 1(a) and Fig. 1(b), respectively.

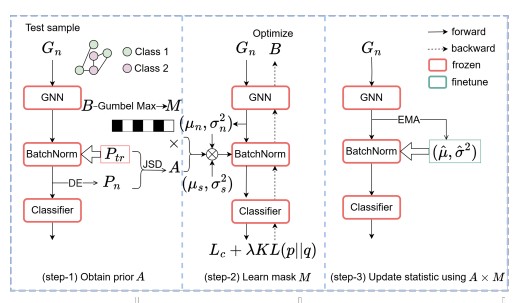 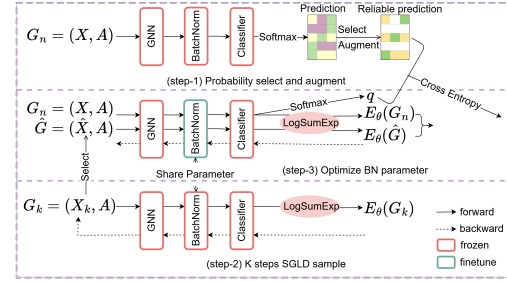

(a) BNSA: optimizing $\mu$ and $\sigma^2$ (Section 3.1)     (b) BNPA: optimizing $\gamma$ and $\beta$ (Section 3.2)

Figure 1: Batch normalization parameter adaptation and batch normalization parameter adaptation

## 3.1 BATCH NORMALIZATION STATISTIC ADAPTATION

BN demonstrates strong empirical performance. However, a comprehensive theoretical understanding of its underlying mechanisms is acknowledged. The original paper (Ioffe and Szegedy, 2015) proposed that BN reduces internal covariate shift, where the distribution of activations changes across layers during training. However, subsequent research, such as Santurkar et al. (Santurkar et al., 2018), has argued that BN works due to its ability to smooth the optimization landscape significantly.

While the theoretical foundation of BN is still under no consensus, we focus on a practical observation: the scale ($\gamma$) and shift ($\beta$) are optimized based on the mean ($\mu$) and variance ($\sigma^2$) during the training stage. If the values of $\mu$ and $\sigma^2$ change significantly on the test data, the batch normalization process using these altered statistics, along with the unchanged $\gamma$ and $\beta$ parameters optimized based on the original $\mu$ and $\sigma^2$, may not perform effectively.

Several works (Li et al., 2017; You et al., 2021; Lim et al., 2023) have verified that a weighted average of the source (training) and target (test) data statistics can improve model inference, as shown in Eq. 2:

$$
\begin{aligned}
\mu &= (1 - \alpha) \cdot \mu_s + \alpha \cdot \mu_t, \\
\sigma^2 &= (1 - \alpha) \cdot \sigma_s^2 + \alpha \cdot \sigma_t^2,
\end{aligned}
\tag{2}
$$

where $(\mu, \sigma^2)$ represents the fine-tuned statistic group of the BN layer, and other parameters in the neural model remain unchanged. $\mu_s$ and $\sigma_s^2$ are estimated from the training data, $\mu_t$ and $\sigma_t^2$ are estimated from the test data, and $\alpha$ is a weighting factor that controls the contribution of each set of statistics.

Most methods (Li et al., 2017; Nado et al., 2020; Schneider et al., 2020; You et al., 2021) determine the value of $\alpha$ empirically, either through experience or grid search. In this paper, we propose a simple yet effective method to determine the value of $\alpha$ by directly calculating the distribution shift between the training and the test data covariates. In Eq. 2, $\alpha$ serves as an indicator of the degree to which the statistic group should favor the test data. If significant differences exist between the covariate distributions, then $\alpha$ should be larger, indicating a stronger reliance on the statistics of test data. Conversely, when differences are minimal, $\alpha$ should be smaller or even zero, favoring the statistics of training data.

As mentioned at the end of Section 1, relying solely on the first and second moments, $(\mu_s, \sigma_s^2)$ and $(\mu_t, \sigma_t^2)$, may not fully capture the distribution shift between training and test data unless the distributions are assumed to be normal or Gaussian. Therefore, we employ non-parametric density estimation to estimate the activation distribution $P_m^{(i,d)}(G_m)$ in the $i$-th BN layer for training instances $G_m \in D_{tr}$, $d$ is the dimensionality of vectors. Similarly, we obtain the estimated distribution $P_n^{(i,d)}(G_n)$ for test instances $G_n \in D_{te}$. The Jensen-Shannon (JS) divergence, a suitable metric for measuring the distance between two probability distributions, is then used to determine the value of $\alpha$, as shown in Eq. 3 and Eq. 4:

$$
JS(P_m^{(i,d)} || P_n^{(i,d)}) = \frac{1}{2} KL(P_m^{(i,d)} || \frac{P_m^{(i,d)} + P_n^{(i,d)}}{2}) + \frac{1}{2} KL(P_n^{(i,d)} || \frac{P_m^{(i,d)} + P_n^{(i,d)}}{2}), \tag{3}
$$

$$\alpha^{(i,d)} = \frac{1}{|D_{tr}| \cdot |D_{te}|} \sum_{n=1}^{|D_{te}|} \sum_{m=1}^{|D_{tr}|} JS(P_m^{(i,d)}||P_n^{(i,d)}). \tag{4}$$

Here, $\alpha^{(i,d)}$ is the weight of dimension $d$ in the $i$-th BN layer, which is the average JS divergence across all pairs of training and test instances.

We can more accurately capture the activation distribution by maintaining an additional distribution of activations alongside the mean ($\mu$) and variance ($\sigma^2$) in BN layers during training. This provides valuable prior information for determining the weighting factor $\alpha$ during TTA. To minimize the computational overhead of maintaining these distributions, we only store the information from the last few training epochs. In all our experiments in Section 4, $P_m^{(i,d)}$ is computed only at the final training epoch. Let $A$ be the matrix storing $\alpha^{(i,d)}$, then $A \in \mathbb{R}^{L \times D}$, where $D$ is the width of the BN layer (i.e., the number of neurons) and $L$ is the number of BN layers in the model architecture.

It is important to note that $\alpha$ is not optimized through a learning process. While the value of $\alpha$ can indicate the relative weighting of training and test data statistics in the BN layer, there is no guarantee that the value of $\alpha$ will necessarily lead to improved performance. To address this issue, we propose two solutions.

First, we propose to learn a mask matrix $M$ to adjust specific dimensions within the BN layer selectively. The mask matrix $M \in \{0, 1\}^{L \times D}$, where $M_{(i,d)} = 1$ indicates that the parameters $(\hat{\mu}_{(i,d)}, \hat{\sigma}^2_{(i,d)})$ in the $i$-th BN layer and $d$-th dimension should be adjusted, and vice versa. We model the mask elements as Bernoulli random variables: $M_{(i,d)} \sim \text{Bernoulli}(b_{(i,d)})$, where $b_{(i,d)}$ are independent Bernoulli variables. To make the sampling process differentiable, we employ the Gumbel-Max trick (Jin et al., 2022a), as shown in Eq. 5, where $\tau$ is a temperature parameter. As $\tau$ approaches 0, the values in $M$ approach binary values.

$$M = sigmoid((\log \delta - \log (1 - \delta) + B)/\tau), \quad \delta \sim U(0, 1). \tag{5}$$

We add the statistic adjustment process into the forward process of BN layers to optimize the Bernoulli variable matrix $B$. In the training process, the modified BN layers are in Eq. 6, where $(\mu_b, \sigma^2_b)$ is the statistics of the test samples in the $b$-th batch. It is worth noting that the statistics of BN layers are not updated during the training process. We integrate the statistic adjustment process into the forward pass of the BN layers, enabling us to optimize the Bernoulli variable matrix $B$. During test-time adaptation (TTA), the modified BN layers operate as shown in Eq. 6:

$$\begin{aligned} \mu &= (1 - (A \odot M)) \cdot \mu_s + (A \odot M) \cdot \mu_b, \\ \sigma^2 &= (1 - (A \odot M)) \cdot \sigma^2_s + (A \odot M) \cdot \sigma^2_b. \end{aligned} \tag{6}$$

Each variable in this equation is implicitly indexed by $(i, d)$, representing the $d$-th dimension in the $i$-th BN layer. The statistics $(\mu_b, \sigma^2_b)$ are computed from the current test batch. Importantly, during TTA, the statistics of the BN layers themselves remain fixed.

We employ contrastive learning for training at test-time adaptation, using the InfoNCE (van den Oord et al., 2019) loss as defined in Eq. 7. In this context, $z_i$ represents the embedding of the $i$-th node, while $\hat{z}_i$ denotes the node embedding after applying DropEdge augmentation (Rong et al., 2020), serving as the positive sample. Negative samples are generated by shuffling the features of nodes (Velickovic et al., 2019), and $\widetilde{z}_i$ represents the corresponding embedding of a negative sample. The cosine similarity between two embeddings is given by $s(z_i, \hat{z}_i) = \frac{z_i^T \hat{z}_i}{||z_i||||\hat{z}_i||}$. Then the loss function is as follows:

$$L_c = \frac{1}{N} \sum_{i=1}^{N} - \log \left( \frac{exp(s(z_i, \hat{z}_i)/\tau)}{exp(s(z_i, \hat{z}_i)/\tau) + exp(s(z_i, \widetilde{z}_i)/\tau)} \right). \tag{7}$$

To preserve the knowledge acquired from the training data, we aim to prevent the updated model from making drastically different predictions compared to the original model. We achieve this by adding the Kullback-Leibler (KL) divergence between the prediction distributions of the two models as a regularization term, as shown in Eq. 8.

$$L_{\text{BNSA}} = L_c + \lambda \cdot KL(p||q), \tag{8}$$

Here, $L_c$ represents the contrastive loss, $\lambda$ is a weighting factor, and $p$ and $q$ denote the prediction distributions of the original and updated models, respectively. By optimizing this loss, we obtain the refined Bernoulli variable matrix $B^*$. This allows us to derive the mask matrix $M^*$, where $M^*_{(i,d)} = 1$ if sigmoid$(b^*_{(i,d)}) > 0.5$, and $M^*_{(i,d)} = 0$ otherwise.

Second, similar to moving average, we further decay the $\mu$ and $\sigma^2$ for $k$ steps as in Eq. 9:

$$
\begin{aligned}
\hat{\mu}_k &= \hat{\mu}_{k-1} \cdot (1 - (A \odot M^*)) + \mu_b \cdot (A \odot M^*), \\
\hat{\sigma}_k^2 &= \hat{\sigma}_{k-1}^2 \cdot (1 - (A \odot M^*)) + \sigma_b^2 \cdot (A \odot M^*),
\end{aligned}
\tag{9}
$$

where $\hat{\mu}_0 = \mu_s, \hat{\sigma}_0^2 = \sigma_s^2$, and $k$ is a small number. Therefore, $\hat{\mu}_k$ and $\hat{\sigma}_k^2$ are the final values of $\mu$ and $\sigma^2$.

## 3.2 BATCH NORMALIZATION PARAMETER ADAPTATION

Once the optimal mean ($\mu$) and variance ($\sigma^2$) are obtained through the BNSA step, we proceed to optimize the parameter group: scale ($\gamma$) and shift ($\beta$). This step further enhances the model's generalization ability and overall performance. However, TTA methods that rely on entropy minimization for parameter optimization can be problematic due to the lack of ground truth labels in the test data. These methods can introduce error signals, leading to increased sensitivity to the learning rate and potential issues like model collapse (Choi et al., 2022; Press et al., 2024).

To alleviate the issue of ground truth forgetting, we propose utilizing the predictions of the pre-trained model before TTA as pseudo-labels for optimizing the scale ($\gamma$) and shift ($\beta$) rather than relying on entropy minimization. However, this approach does not address the challenge well, as the pseudo-labels may be inaccurate and lack the guarantee of ground truth.

If the pseudo-labels are largely accurate, they can be confidently used for parameter optimization. To leverage the model's confidence across all classes, we propose utilizing soft labels, i.e., the entire softmax output, rather than hard labels (the class with the highest probability). To ensure the selection of high-quality soft pseudo-labels, we employ two strategies:

1. Entropy-based Selection: We select test instances with low entropy in their softmax outputs. This focuses on instances where the model exhibits high confidence in its predictions.

2. Confidence-based Filtering: We retain only the high and low probabilities within the softmax output, discarding intermediate values. This preserves the model's discriminative ability for classes in which it demonstrates confidence while filtering out less certain predictions.

However, an implicit assumption underlies these two strategies: the model should be well-calibrated. A calibrated model refers to a model whose predicted probabilities align with the observed frequencies of the classes. In other words, for a well-calibrated model, if a class is predicted with a probability of 0.8, we would expect that class to be the true label approximately 80% of the time. Inspired by the concept of energy-based models (EBMs) applied to classifiers (Grathwohl et al., 2020), we incorporate an EBM into our TTA framework to optimize the BN layer's parameter group (scale and shift). This integration aims to enhance both the model's generalization ability and its calibration. The joint energy-based model (JEM) is defined in Eq. 10:

$$
\log p_\theta(x, y) = \log p_\theta(x) + \log p_\theta(y|x),
\tag{10}
$$

where $x$ represents the node representation and $y$ denotes the corresponding label.

We follow the classic formulation of EBMs (LeCun et al., 2006): $p_\theta(x) = \frac{\exp(-E_\theta(x))}{Z(\theta)}$, where $E_\theta(x)$ is the energy function and $Z(\theta)$ is the partition function. The energy function is defined as:

$$
E_\theta(x) = -\text{LogSumExp}_y(f_\theta(x)[y]) = -\log \sum_y \exp(f_\theta(x)[y]).
\tag{11}
$$

During optimization, the derivative of $\log p_\theta(x)$ in Eq. 10 can be rewritten as:

$$
\frac{\partial \log p_\theta(x)}{\partial \theta} = \mathbb{E}_{p_\theta(x')}\left[\frac{\partial E_\theta(x')}{\partial \theta}\right] - \frac{\partial E_\theta(x)}{\partial \theta}.
\tag{12}
$$

The expectation in Eq. 12 requires sampling. We utilize Stochastic Gradient Langevin Dynamics (SGLD) (Welling and Teh, 2011) to approximate this sampling process, as shown in Eq. 13:

$$\hat{x}_0 \sim p_0(x), \quad \hat{x}_{i+1} = \hat{x}_i - \frac{\delta}{2} \frac{\partial E_\theta(\hat{x}_i)}{\partial \hat{x}_i} + \epsilon, \quad \epsilon \sim \mathcal{N}(0, \delta), \tag{13}$$

where $p_0(x)$ is a uniform distribution, $\delta$ is the step size, and $\epsilon$ is Gaussian noise. To reduce the computational cost, we approximate the right-hand side of Eq. 12 using Persistent Contrastive Divergence (PCD) (Tieleman, 2008). The approximated generation loss $L_{gen}$ is given by:

$$L_{gen} = \log p_\theta(x) = \text{LogSumExp}_{y'}(f_\theta(x)[y']) - \text{LogSumExp}_{y'}(f_\theta(\hat{x})[y']), \tag{14}$$

where $y'$ denotes the pseudo-labels. Sampling is performed for $T$ steps. The sample adopted is the one whose energy is the closest to the real one in the sampling process, i.e., $\hat{x} = \arg\min(E_\theta(x) - E_\theta(\hat{x}_i))$.

The conditional probability $\log p_\theta(y|x)$ in Eq. 10 is modeled using cross-entropy. As previously mentioned, we employ entropy-based selection and confidence-based filtering to include reliable test instances and pseudo-labels. We use the model before adaptation to obtain the softmax predictions $p$, which serve as our pseudo-labels. Let $i$ be the node index, then the entropy of node $i$ is calculated as follows: $Entropy(p^{(i)}) = -\sum_{c=1}^{C} p_c^{(i)} \log p_c^{(i)}$, where $c$ represents a class label. The score $s^{(i)}$ of node $i$ is then defined as:

$$s^{(i)} = \mathbb{I}(Entropy(p^{(i)}) < \tau_e), \tag{15}$$

where $\mathbb{I}$ is the indicator function, and $\tau_e$ is a hyperparameter that acts as an entropy threshold.

Furthermore, we retain only the high and low probabilities within the softmax output, discarding intermediate values to enhance the model's discriminative ability. The weight $w_c^{(i)}$ assigned to node $i$ for class $c$ is defined as:

$$w_c^{(i)} = \exp\left(p_c^{(i)} - \frac{\tau_c^1 + \tau_c^2}{2}\right) \cdot \mathbb{I}(p_c^{(i)} \geq \tau_c^1 | p_c^{(i)} \leq \tau_c^2). \tag{16}$$

$\tau_c^1$ and $\tau_c^2$ are probability thresholds. The final classification loss $L_{clf}$ is then given by:

$$L_{clf} = p_\theta(y|x) = -\frac{1}{\sum s^{(i)}} \sum_{i=1}^{N} \sum_{c=1}^{C} s^{(i)} \cdot w_c^{(i)} \cdot p_c^{(i)} \cdot \log q_c^{(i)}, \tag{17}$$

where $w_c^{(i)}$ is the weight assigned to node $i$ for class $c$, and $q$ represents the updated softmax output after adaptation. The overall loss function for BNPA is

$$L_{\text{BNPA}} = L_{gen} + L_{clf}. \tag{18}$$

While our proposed BNPA method offers improved performance, it's important to acknowledge the potential increase in computational time compared to methods that rely on entropy minimization for parameter optimization. The primary factor contributing to this increase is the SGLD sampling process. However, this additional computational cost is often offset by significant performance gains and stable adaptation, especially in scenarios where accuracy is foremost.

## 4 EXPERIMENTAL EVALUATION

We conducted experiments on seven datasets: Amazon-Photo (Shchur et al., 2018), Cora (Yang et al., 2016), Twitch-E (Rozemberczki et al., 2021), FB-100 (Traud et al., 2012), OGB-Products (Hu et al., 2020), Elliptic (Pareja et al., 2020), and OGB-Arxiv (Hu et al., 2020). These datasets encompass various types of distribution shifts, both synthetic and natural. To ensure a fair comparison of the effectiveness in handling the distribution shift and mitigating performance degradation, we evaluate all methods under the same out-of-distribution (OOD) setting, which is also employed in GTRANS (Jin et al., 2022b) and EERM (Wu et al., 2022). Detailed dataset and OOD descriptions are provided in Appendix A.2.

**Baselines.** We compare our proposed method with seven state-of-the-art (SOTA) test-time adaptation (TTA) methods, categorized as follows:

Table 1: Mean test dataset accuracy (%) for node classification. All results are averaged over ten runs with random seeds. Bold and underlined indicate the best and second-best results, respectively.

| Backbone | Method | Amazon-Photo | Cora | Elliptic | FB-100 | OGB-Arxiv | OGB-Products | Twitch-E | Rank |
|---|---|---|---|---|---|---|---|---|---|
| GCN | a-BN(You et al., 2021) | 94.98±0.57 | 95.37±0.61 | 53.41±2.50 | 53.15±0.70 | 51.89±0.36 | 61.26±0.11 | 60.34±0.54 | 4.1 |
| | DUA(Mirza et al., 2022) | 94.65±0.61 | 94.00±0.71 | 57.24±2.72 | 53.05±0.84 | 52.77±0.33 | 61.27±0.11 | 60.03±0.56 | 4.9 |
| | MEMO(Zhang et al., 2022) | 91.56±0.66 | 89.67±0.76 | 58.66±3.02 | 53.17±1.09 | 52.12±0.33 | 58.52±0.86 | 60.27±0.50 | 5.9 |
| | TENT(Wang et al., 2021) | 94.03±1.07 | 91.87±1.36 | 51.71±2.00 | 54.16±1.00 | 45.72±0.67 | 60.69±0.15 | 59.46±0.55 | 6.4 |
| | SAR(Niu et al., 2023) | 94.59±0.63 | 93.46±0.75 | 50.75±2.10 | 53.22±0.86 | 52.44±0.33 | 61.28±0.11 | 60.09±0.56 | 4.7 |
| | DELTA(Zhao et al., 2023) | 94.67±0.59 | 94.03±0.71 | 62.77±1.80 | 53.07±0.84 | 52.78±0.33 | 61.23±0.11 | 60.05±0.55 | 4.1 |
| | GTRANS(Jin et al., 2022b) | 94.13±0.77 | 94.66±0.63 | 55.88±3.10 | 54.32±0.60 | 50.18±0.63 | 60.64±0.13 | 60.42±0.86 | 4.4 |
| | Ours | 96.17±0.23 | 98.21±0.42 | 64.16±1.28 | 53.19±0.84 | 52.90±0.28 | 61.29±0.13 | 60.47±0.61 | 1.4 |
| GraphSAGE | a-BN(You et al., 2021) | 97.44±0.54 | 99.96±0.02 | 60.95±4.08 | 53.70±0.39 | 51.55±0.19 | 63.85±0.13 | 62.17±0.14 | 5.0 |
| | DUA(Mirza et al., 2022) | 95.57±0.57 | 99.89±0.05 | 63.54±3.13 | 53.73±0.37 | 54.33±0.14 | 63.91±0.11 | 62.38±0.15 | 4.3 |
| | MEMO(Zhang et al., 2022) | 96.31±0.79 | 99.94±0.04 | 58.05±5.12 | 53.16±0.55 | 53.91±0.22 | 63.86±0.11 | 62.39±0.12 | 4.7 |
| | TENT(Wang et al., 2021) | 95.72±0.43 | 99.80±0.10 | 55.89±4.87 | 54.86±0.34 | 48.07±0.44 | 62.81±0.16 | 62.09±0.09 | 6.3 |
| | SAR(Niu et al., 2023) | 95.11±0.57 | 99.75±0.12 | 57.91±4.24 | 53.85±0.37 | 54.01±0.17 | 63.90±0.12 | 62.36±0.16 | 5.4 |
| | DELTA(Zhao et al., 2023) | 95.60±0.57 | 99.90±0.05 | 65.93±2.27 | 53.75±0.36 | 54.34±0.14 | 63.88±0.14 | 62.39±0.15 | 3.7 |
| | GTRANS(Jin et al., 2022b) | 96.91±0.68 | 99.45±0.13 | 60.81±5.19 | 54.64±0.62 | 52.99±0.28 | 64.17±0.22 | 62.15±0.13 | 4.6 |
| | Ours | 99.39±0.26 | 99.99±0.01 | 68.53±1.46 | 54.01±0.34 | 54.41±0.17 | 63.85±0.11 | 62.41±0.14 | 2.0 |
| GAT | a-BN(You et al., 2021) | 96.62±0.55 | 98.40±0.44 | 65.93±2.07 | 50.07±1.12 | 53.81±0.47 | 67.43±0.14 | 58.38±1.83 | 3.1 |
| | DUA(Mirza et al., 2022) | 96.14±0.67 | 97.45±0.82 | 65.75±2.23 | 49.93±1.13 | 53.83±0.49 | 67.30±0.14 | 58.03±1.67 | 5.4 |
| | MEMO(Zhang et al., 2022) | 95.54±1.06 | 97.83±0.45 | 58.55±5.40 | 49.76±0.71 | 53.96±0.47 | 66.90±0.34 | 57.84±1.94 | 6.4 |
| | TENT(Wang et al., 2021) | 95.99±0.46 | 95.91±1.14 | 66.07±1.66 | 51.47±1.70 | 50.87±0.23 | 66.03±0.47 | 58.33±1.18 | 5.3 |
| | SAR(Niu et al., 2023) | 95.99±0.72 | 95.72±1.37 | 64.47±2.41 | 50.06±1.28 | 54.09±0.48 | 67.42±0.16 | 58.11±1.67 | 5.3 |
| | DELTA(Zhao et al., 2023) | 96.14±0.66 | 97.53±0.81 | 66.42±1.87 | 49.93±1.14 | 53.19±0.52 | 67.37±0.17 | 58.11±1.68 | 4.9 |
| | GTRANS(Jin et al., 2022b) | 96.67±0.74 | 96.37±1.00 | 66.43±2.57 | 51.16±1.72 | 52.59±0.66 | 67.27±0.14 | 58.59±1.07 | 3.7 |
| | Ours | 97.09±0.52 | 99.63±0.10 | 68.03±1.60 | 50.07±1.33 | 54.14±0.54 | 67.44±0.14 | 58.13±1.63 | 1.9 |

- BN statistic modification. a-BN (You et al., 2021) and DUA (Mirza et al., 2022) maintain statistics from both training and test instances, utilizing a weighted summarization for batch normalization.

- Model parameter optimization. TENT (Wang et al., 2021) fine-tunes parameters within BN layers using entropy minimization. SAR (Niu et al., 2023) selectively fine-tunes parameters on a subset of test samples using entropy minimization. DELTA (Zhao et al., 2023) incorporates class-specific weights into the loss calculation during fine-tuning. MEMO (Zhang et al., 2022) fine-tunes all model parameters by minimizing entropy across augmented data.

- Input augmentation. GTRANS (Jin et al., 2022b) learns to augment input graphs to better align with the model without fine-tuning model parameters.

Implementation details are provided in Appendix A.3. Experimental settings for baseline methods were adopted from their respective publications. All reported results represent the average performance over ten independent runs with different random seeds. We evaluate our method across three commonly used GNN backbone models: GCN (Kipf and Welling, 2016), GraphSAGE (Hamilton et al., 2017), and GAT (Velickovic et al., 2018).

## 4.1 RESULTS

Table 1 presents the classification performance. Our proposed algorithm consistently achieves strong out-of-distribution generalization ability across diverse datasets, achieving the best or near-best results on most datasets. In particular, our algorithm achieves the highest classification accuracy on GCN across six datasets (Amazon-Photo, Cora, Elliptic, OGB-Arxiv, OGB-Products, and Twitch-E), with significant improvements over the second-best algorithm on Amazon-Photo, Cora, and Elliptic (1.19%, 2.84%, and 1.39%, respectively). On GraphSAGE, our method yields the best results on five datasets (Amazon-Photo, Cora, Elliptic, OGB-Arxiv, and Twitch-E), with notable gains on Amazon-Photo and Elliptic (1.95% and 2.60%). While not the top performer on OGB-Products, our algorithm remains highly competitive. Using GAT, our algorithm achieves the best performance on five datasets (Amazon-Photo, Cora, Elliptic, OGB-Arxiv, and OGB-Products), with solid improvements on Cora and Elliptic (1.23% and 1.60%). Performance on the remaining datasets is either excellent or very close to the best results.

Table 2: Ablation Study. Bold and underlined denote the best and second-best results, respectively.

| Backbone | Method | Amazon-Photo | Cora | Elliptic | FB-100 | OGB-Arxiv | OGB-Products | Twitch-E |
|---|---|---|---|---|---|---|---|---|
| | BNSA | 96.08 | 97.72 | 60.52 | 54.10 | 51.95 | 60.92 | 58.94 |
| | BNSA w/o A | 95.64 | 94.54 | 52.75 | 54.00 | 51.34 | 60.63 | 58.74 |
| | BNSA w/o M | 96.06 | 97.69 | 60.07 | 53.53 | 51.18 | 60.84 | 58.83 |
| GCN | BNPA | 95.51 | 91.81 | 49.21 | 54.13 | 52.50 | 61.11 | 58.69 |
| | BNPA w/o MinSamp | 95.46 | 91.53 | 49.21 | 54.12 | 51.78 | 61.09 | 58.69 |
| | BNPA w/o CEselc | 95.45 | 91.46 | 48.95 | 54.13 | 51.52 | 60.99 | 58.62 |
| | Overall | 96.45 | 98.09 | 61.09 | 54.14 | 52.68 | 61.26 | 59.16 |

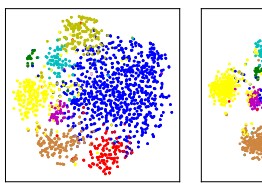

(a) Cora G5 BNPA (b) Cora G5 Overall

Figure 2: t-SNE with and without BNPA

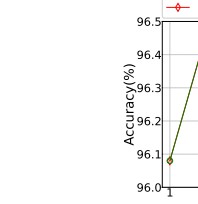
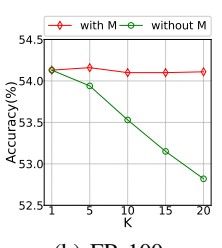

(a) Amazon-Photo (b) FB-100

Figure 3: Mask matrix $M$ and hyperparameter $k$

## 4.2 ABLATION STUDIES

An ablation study was conducted on GCN (see Table 2) to analyze the impact of individual components in our proposed TTA method. Different settings were tested, including using only BN statistic adaptation (BNSA) or BN parameter adaptation (BNPA), variations of BNSA without the adjustment weight $A$ (BNSA w/o $A$) or mask matrix $M$ (BNSA w/o $M$), variations of BNPA without the closest sample selection (BNPA w/o MinSamp) or the entropy and confidence-based selection (BNPA w/o CEselc).

Results highlight the importance of both BNSA and BNPA, as well as the effectiveness of our proposed weighting and selection strategies. Using a fixed weight instead of our proposed distribution shift-based approach negatively impacts performance, validating our weighting strategy. Removing the mask matrix $M$, which selectively adjusts BN statistics, also decreases performance. Minor but consistent performance decreases are observed when the closest sample in SGLD is not selected, or the entropy and confidence-based selection strategy is removed. All highlight the importance of these components. While the overall improvement of BNSA over BNSA w/o $M$ might seem marginal in some cases, the mask's effectiveness extends beyond accuracy improvements. It also contributes to enhanced model calibration, as Appendix A.4.4 demonstrates. This improved calibration leads to more reliable confidence estimates, which are essential for various downstream tasks.

To further understand the impact of BNSA, we visualize the activations from the last BN layer when using BNPA alone and in conjunction with BNSA (Fig. 2). The results indicate that solely utilizing BNPA does not yield the same level of class separation as the combined approach. Incorporating both BNSA and BNPA leads to more distinct and inherent representations for each class.

## 4.3 HYPERPARAMETER SENSITIVITY STUDIES

We highlight the effectiveness of the mask matrix $M$ in our approach. Fig. 3 illustrates the stability of $M$ across learning rounds, demonstrating its limited impact on Amazon-Photo but significant contribution to preventing model collapse in FB-100. Additional results are presented in the Appendix.

Fig. 3 demonstrates that our proposed method achieves strong performance even with a small value of $k$, aligning with practical considerations of preserving information from the training data. Fig. 4 illustrates the difference between the mean and variance calculated in Eq. 9 and those maintained from the training data. This difference highlights how our method effectively balances the influence of statistics from both the training and test datasets during adaptation.

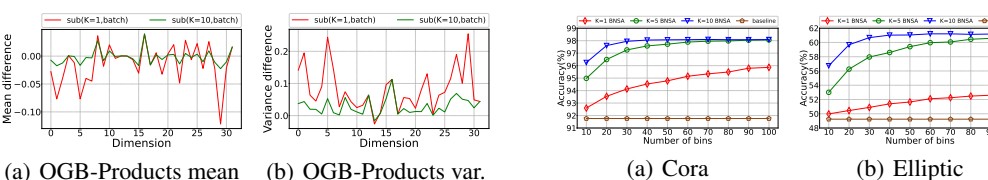

| (a) OGB-Products mean | (b) OGB-Products var. | (a) Cora | (b) Elliptic |
|---|---|---|---|

Figure 4: Mean's and variance's difference          Figure 5: # of bins in BNSA vs accuracy

We investigate the impact of histogram size in non-parametric density estimation in Fig. 5. Results show that: (1) BNSA consistently improves performance when the number of bins is small; (2) accuracy eventually stabilizes with increasing bins beyond 30.

## 5    CONCLUSION

This work presented a novel two-step approach for test-time adaptation (TTA) of graph neural networks. Our method adapts batch normalization statistics to the test data and refines model parameters using an energy-based model. This approach addresses the limitations of existing methods, improving model generalization and calibration. Empirical evaluation across seven diverse datasets demonstrates its superior performance compared to state-of-the-art TTA techniques.

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

# A    APPENDIX / SUPPLEMENTAL MATERIAL

## A.1    RELATED WORK

Test-time adaptation aims to bridge the gap between a model's training data and the test data of potential data distribution shift, thereby mitigating performance degradation. Usually, in TTA, only unlabeled test data and a pre-trained neural model are available. Existing approaches to fully TTA can be clarified into two primary categories: batch normalization calibration and model optimization.

### A.1.1    BATCH NORMALIZATION CALIBRATION

BN layers in neural networks typically use statistics of mean and variance calculated from the training data to normalize activations during inference. However, this can be problematic when the test data distribution differs significantly from the training data.

Several methods have been proposed to address this issue: AdaBN (Li et al., 2017) replaces source statistics with estimates from the entire target domain, arguing that BN statistics can encode domain-specific information. PredBN (Nado et al., 2020) utilizes current batch statistics for normalization, and PredBN+ (Schneider et al., 2020) combines source and current batch statistics based on batch size. $\alpha$-BN (You et al., 2021) uses a manually defined hyperparameter to blend source and target statistics. DUA (Mirza et al., 2022) updates BN statistics using a decaying momentum and exponential moving average on the target dataset.

These methods often rely on heuristic weights for combining statistics, limiting their flexibility and stability. In contrast, our proposed method dynamically adjusts BN statistics using weights based on the magnitude of the distribution shift and a learned mask matrix, enhancing performance on target data with distribution shift.

### A.1.2    MODEL OPTIMIZATION

The other category of TTA techniques focuses on optimizing the pre-trained model directly on the test dataset, using an articulated-designed, unsupervised objective. Methods to optimize the whole model are as follows. MEMO (Zhang et al., 2022) minimizes the entropy of average predictions across augmented views of test samples. CoTTA (Wang et al., 2022) minimizes entropy based on pseudo-labels derived from weighted and augmentation averaging. AdaContrast (Chen et al., 2022a) employs self-supervised contrastive learning with pseudo-labeling. GAPGC (Chen et al., 2022b) adapts contrastive learning to graph neural networks.

Methods to optimize the partial parameters or layers are as follows: TENT (Wang et al., 2021) minimizes entropy of predictions on test samples to optimize BN parameters. HLR (Mummadi et al., 2021) uses an unsaturated proxy loss and discrepancy regularization to optimize BN parameters. T3A (Iwasawa and Matsuo, 2021) adjusts the classifier using pseudo-prototype representations. PCL (Su et al., 2023) optimizes inter-layer normalization parameters using perturbations in the feature space.

These methods may face challenges when the pre-trained model performs poorly on the target data, as unsupervised objectives like entropy minimization or contrastive learning may be less effective without access to ground truth labels. To address this, our proposed approach leverages an energy-based joint model training approach, exploiting the benefits of generative models to select and enhance the initial model's predictions simultaneously. This leads to more reliable adapted predictions.

## A.2    SETUP

To ensure a fair comparison of the effectiveness in handling the distribution shift and mitigating performance degradation, we evaluate all methods following the same experimental settings in two recent works, GTRANS (Jin et al., 2022b) and EERM (Wu et al., 2022). We conducted experiments on seven datasets on three GNN backbone models.

**datasets** The datasets contain various distribution shift types, synthetic and natural. According to different shift types, we categorize the seven datasets as follows:

- Synthetic shift. In Amazon-Photo (Shchur et al., 2018) and Cora (Yang et al., 2016), the synthetic shift is involved by adding artificial node features. The added node features are

different in each graph, so when the dataset is divided into training, validation, and test samples, they contain different synthetic shifts, which are used to evaluate the model's generalization ability. Distribution shifts are introduced into the training and testing data to evaluate the model's ability for out-of-distribution generalization. We use the provided node features for each dataset to construct node labels and spurious environment-sensitive features.

- Domain shift. In Twitch-E (Rozemberczki et al., 2021), FB-100 (Traud et al., 2012), and OGB-Products (Hu et al., 2020), there are domain shifts. The nodes in different graphs are from different domains. The implications of such domain shifts are profound, impacting the generalizability and performance of models trained on these datasets. The training, validation, and test samples are from different domains.

- Temporal shift. Dataset Elliptic (Pareja et al., 2020) and OGB-Arxiv (Hu et al., 2020) contain temporal shifts; graph nodes originate from different periods. This temporal variation implies that the data used for the training, validation, and testing phases are not homogenous with respect to time. So, training, validation, and test samples are from different periods.

For Amazon-Photo, Cora, Elliptic, FB-100, and Twitch-E, we use the same partition ratio as the one in GTRANS. Specifically, Amazon-Photo and Cora have 1/1/8 graphs for training/validation/test sets. Twitch-E has 1/1/5 graphs, FB-100 has 3/2/3 graphs, and Elliptic has 5/5/33 graphs for training/validation/test sets.

For OGB-Arxiv, more than one-fourth class in the training samples in GTRANS contains only a small number of nodes, which we consider unfair because this is dropping information by purpose. So, we use the suggested partition ratio in the OGB standard for OGB-Arxiv and OGB-Product, which is 3/1/1 for OGB-Arxiv and OGB-Products.

### A.3 IMPLEMENTATION DETAILS

During model training on the training set, the number of layers was set to 5 for Elliptic and OGB-Products and 2 for other datasets. The GAT model had four attention heads per layer, and the representation dimension was set to 32 for all datasets. BN layers were included in all model architectures. The learning rate was set to 0.001 for Amazon-Photo and Cora and 0.01 for other datasets. The number of training epochs was set to 500 for OGB-Arxiv and 200 for other datasets.

Our proposed algorithm utilized histogram density estimation for non-parametric density estimation. The number of bins was set to 100 for Amazon-Photo, Cora, and Elliptic. For Twitch-E, the number of bins was 100 for GCN and 10 for GraphSAGE and GAT. For other datasets, ten bins were used.

During the learning of the mask matrix $M$ for adjusting weights, the loss weight $\lambda$, learning rate, and number of training epochs were set to [0.1, 0.01, 300] for Amazon-Photo and Cora, [1.5, 0.2, 300] for OGB-Arxiv, [0.8, 0.1, 300] for GCN and GraphSAGE on FB-100, [0.8, 0.1, 10] for GAT on FB-100, [0.1, 0.1, 300] for GCN on Twitch-E, [0.8, 0.1, 300] for GraphSAGE and GAT on Twitch-E, [0.1, 0.01, 300] for GCN and GraphSAGE on Elliptic, [0.1, 0.01, 100] for GAT on Elliptic, and [0.8, 0.1, 300] for OGB-Products.

In the BN statistics adaptation process, the number of adaptation rounds was set to 10 for Amazon-Photo, Cora, and Elliptic. For Twitch-E, it was 10 for GCN and 1 for GraphSAGE and GAT. For other datasets, one adaptation round was used.

During BN parameter optimization, the learning rate and the number of optimization rounds were set to [0.0001, 10] for GCN and GAT on Amazon-Photo, [0.001, 80] for GraphSAGE on Amazon-Photo, [0.0001, 10] for Cora, Twitch-E, Elliptic, and OGB-Products, [0.001, 10] for GCN on OGB-Arxiv, [0.0001, 10] for GraphSAGE on OGB-Arxiv, [0.001, 20] for GAT on OGB-Arxiv, [0.001, 30] for GCN on FB-100, and [0.0001, 10] for GraphSAGE and GAT on FB-100. In the SGLD sampling process, step size $\delta$ is set to 2.0, and the number of steps $T$ is 30. In the process of entropy-based selection and confidence-based filtering, for flexibility and versatility, entropy threshold $\tau_e$ is determined based on the value that lies at the scale boundary after sorting, We keep the nodes whose entropy is in the top 60%, for confidence threshold $\tau_c^1$, $\tau_c^2$, We set it uniformly as [0.2, 0.8].

## A.4 ADDITIONAL RESULTS

### A.4.1 T-SNE WITH AND WITHOUT BNPA

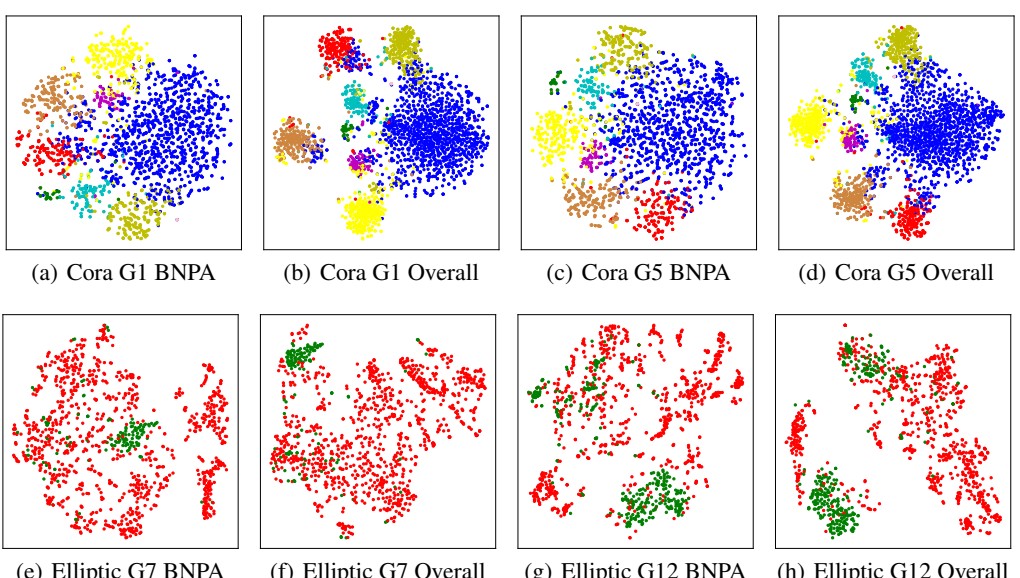

| (a) Cora G1 BNPA | (b) Cora G1 Overall | (c) Cora G5 BNPA | (d) Cora G5 Overall |
| --- | --- | --- | --- |

| (e) Elliptic G7 BNPA | (f) Elliptic G7 Overall | (g) Elliptic G12 BNPA | (h) Elliptic G12 Overall |
| --- | --- | --- | --- |

Figure 6: t-SNE with and without BNPA

To further understand the impact of BNSA, we visualize the activations from the last BN layer when using BNPA alone and in conjunction with BNSA (Fig. 6) on two data sets. The results indicate that solely utilizing BNPA does not yield the same level of class separation as the combined approach. Incorporating both BNSA and BNPA leads to more distinct and inherent representations for each class.

### A.4.2 MASK MATRIX M AND HYPERPARAMETER K

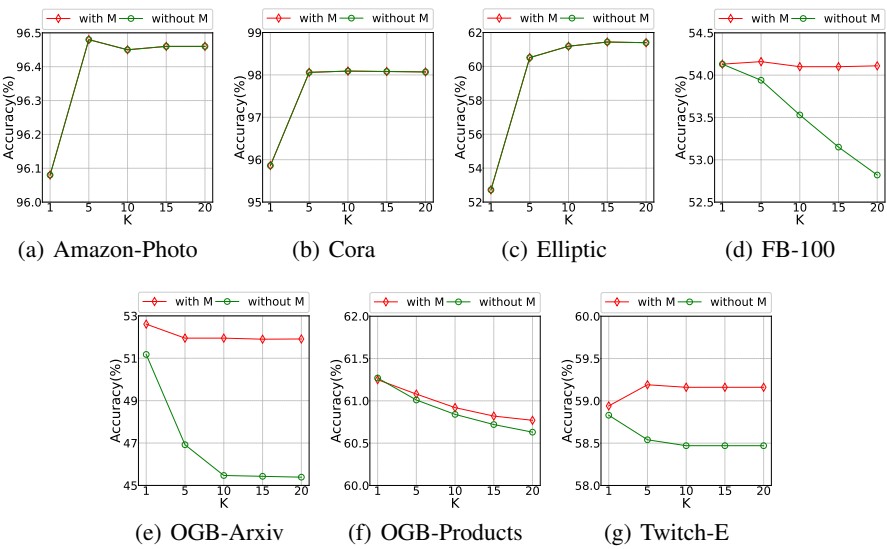

| (a) Amazon-Photo | (b) Cora | (c) Elliptic | (d) FB-100 |
| --- | --- | --- | --- |

| (e) OGB-Arxiv | (f) OGB-Products | (g) Twitch-E |
| --- | --- | --- |

Figure 7: Mask matrix $M$ and hyperparameter $k$

We highlight the effectiveness of the mask matrix $M$ in our approach. Fig. 7 illustrates the stability of $M$ across learning rounds, demonstrating its limited impact on Amazon-Photo but significant contribution to preventing model collapse.

### A.4.3 MEAN'S AND VARIANCE'S DIFFERENCE

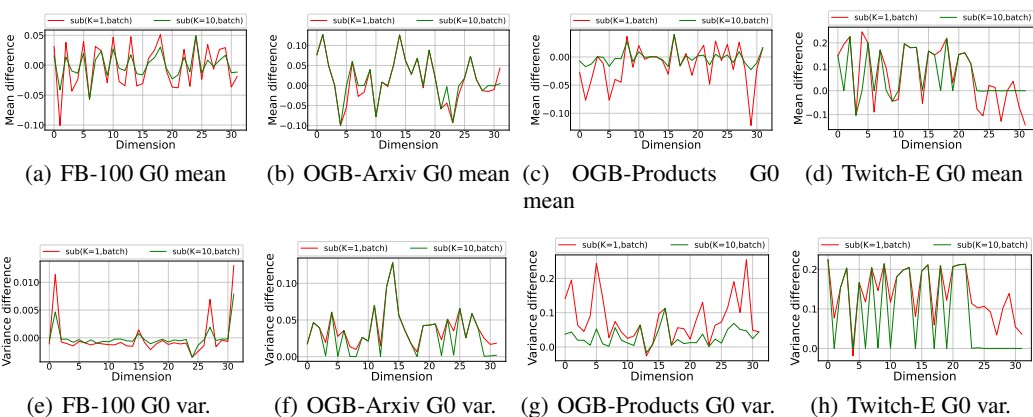

(a) FB-100 G0 mean  (b) OGB-Arxiv G0 mean  (c) OGB-Products G0 mean  (d) Twitch-E G0 mean

(e) FB-100 G0 var.  (f) OGB-Arxiv G0 var.  (g) OGB-Products G0 var.  (h) Twitch-E G0 var.

Figure 8: Mean's and variance's difference

Fig. 7 demonstrates that our proposed method achieves strong performance even with a small value of $k$, aligning with practical considerations of preserving information from the training data. Fig. 8 illustrates the difference between the mean and variance calculated in Eq. 9 and those maintained from the training data. This difference highlights how our method effectively balances the influence of statistics from both the training and test datasets during adaptation.

### A.4.4 THE MODEL CALIBRATION BY BNSA

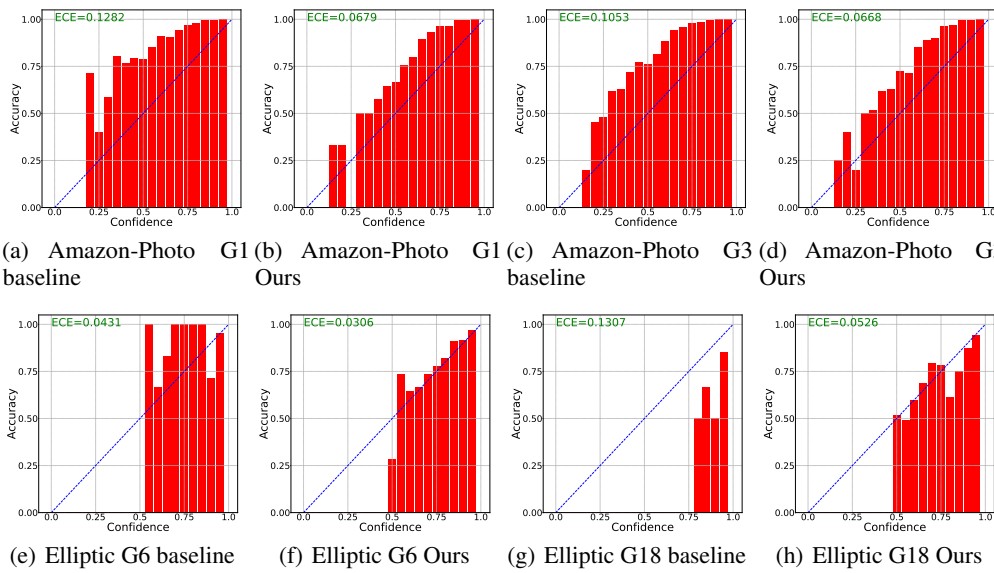

(a) Amazon-Photo G1 baseline  (b) Amazon-Photo G1 Ours  (c) Amazon-Photo G3 baseline  (d) Amazon-Photo G3 Ours

(e) Elliptic G6 baseline  (f) Elliptic G6 Ours  (g) Elliptic G18 baseline  (h) Elliptic G18 Ours

Figure 9: The model calibration by BNSA

A well-calibrated classifier exhibits predictive confidence that aligns with its misclassification rate. As discussed in Section 3.2, incorporating an energy-based model (EBM) can enhance model calibration, leading to improved pseudo-label selection. To demonstrate this, we present calibration plots for all

datasets, comparing the baseline (no adaptation) and our proposed approach (BNSA + BNPA). The baseline model shows poor calibration, while our method effectively calibrates the GNN model. This improvement is evident in the Expected Calibration Error (ECE) values, which quantify the average discrepancy between classifier confidence and accuracy.

## A.5 ALGORITHM

The pseudo-code for Algorithms BNSA and BNPA is presented in this section.

---

**Algorithm 1** BNSA

**Input:** A test graph $G_n \in D_{te}$, trained model $f_\theta$ include $L$ BN layers with BN statistic $(\mu_s, \sigma_s^2)$, learn mask epochs $T$, temperature parameter $\tau$.
**Output:** updated statistic $(\hat{\mu}, \hat{\sigma}^2)$.
1: **for** $i = 1, ..., L$ **do**
2:    Represent the distribution $P_n^{(i)}$ using non-parametric density estimation;
3: **end for**
4: Compute the weight $a^{(i,d)}$ using Eq. 3 and 4;
5: Prior weights $A = [a^{(0)}, a^{(1)}, ..., a^{(L)}]$;
6: **for** $t = 0, ..., T - 1$ **do**
7:    $M = sigmoid((\log \delta - \log (1 - \delta) + B)/\tau), \delta \sim U(0,1)$;
8:    Update $B$ using loss as shown in Eq. 8 and BN statistic in Eq. 6;
9: **end for**
10: **if** $sigmoid(B^*) > 0.5$ **then**
11:    $M^* = 1$;
12: **else**
13:    $M^* = 0$;
14: **end if**
15: Adapt the statistic using Eq. 9;
    **return** $(\hat{\mu}, \hat{\sigma}^2)$.

---

**Algorithm 2** BNPA

**Input:** A test graph $G_n \in D_{te}$, trained model $f_\theta$ with BN parameter $(\beta, \gamma)$, learn epochs $K$, replay buffer $B_f$, reinitialization frequency $p_{ri}$.
**Output:** updated BN parameter $(\beta^*, \gamma^*)$.
    **for** $k = 0, ..., K - 1$ **do**
2:    Model forward with $G_n$;
    Compute $L_{clf}$ using Eq. 17;
4:    Sample $\hat{x}_0 \sim B_f$ with probability $1 - p_{ri}$, else $\hat{x}_0 \sim U(-1,1)$;
    Obtain the sample $\hat{x}$ using Eq. 13;
6:    Add $\hat{x}$ to $B_f$;
    Compute $L_{gen}$ using Eq. 14;
8:    Total loss is $L_{clf} + L_{gen}$;
    Backward and optimize $(\beta, \gamma)$;
10: **end for**
    **return** $(\beta^*, \gamma^*)$.

---

