# OpenReview forum: "Optimizing Activations Beyond Entropy Minimization for Test-Time Adaptation of Graph Neural Networks"
_ICLR.cc/2025/Conference — Submitted to ICLR 2025_

### Official Review · Reviewer_qvGx · 2024-10-27

**Soundness:** 3
**Presentation:** 3
**Contribution:** 3
**Rating:** 6
**Confidence:** 1

**Summary:**

The paper offers a method to optimize different batch normalization parameters when performing test-time adaptations for GNNs. The paper focuses on optimizing all the BN parameters rather than just the mean and variance, and shows empirically that the proposed method outperform the previous such method in a significant margin across several datasets using number of architectures.

**Strengths:**

Originality: the paper suggests learning other parameters of BN layers, which makes sense given the methods suggested in previous works. As far as I can tell, this method is original and novel, and well places in the current literature.

Clarity: The paper is written in a clear way, and it was easy to follow the suggested ideas and the motivations behind them.

Quality: The suggested method is tested across several benchmarks using different architectures. I'm fairly convinced that the method work, and improves the upon the previous methods. The papers presents a good ablation study which checks the different parts of the method.

Significance: The results presented show notable improvements over previous works, suggesting that the approach could positively impact the field.

**Weaknesses:**

Originality: While the method is novel, it may be perceived as somewhat incremental, as it only modifies BN layers, similar to several previous studies.

Quality: The paper could be strengthened with additional results, particularly a comparison of computation times with baseline methods. Since the approach calculates second-order statistics, this might incur computational costs, which would be useful to discuss. Additionally, as the baseline methods also approximate the mean and variance of the BN layers, it would be helpful to see if using the suggested method to calculate only the scale and shift of the BN layers, but then using the mean and variance calculated by the baseline methods is also beneficial.

Significance: The scalability of this method to larger networks and datasets is unclear. Without the computational cost analysis and baseline comparisons, it’s challenging to fully assess the impact of the results.

----

Summary:

The paper could be significantly improved by including the suggested additional experiments. Nonetheless, I believe it has merit for publication, which is why I recommend a weak accept.

Disclaimer: This paper is outside my primary area of expertise. My review provides only a general assessment, and there may be aspects regarding related work or specific methodological details that I have missed.

**Questions:**

Can the method be implemented when the data is significantly larger? How fast it is to implement it?

What happens if you use the mean and variance calculated by other methods, and calculate the scale and shift using your method?

---

### Official Review · Reviewer_xPKS · 2024-10-28

**Soundness:** 2
**Presentation:** 2
**Contribution:** 2
**Rating:** 3
**Confidence:** 4

**Summary:**

This paper presents a new approach for test-time adaptation (TTA) in classification models, specifically targeting graph neural networks (GNNs). This work optimizes the activations in batch normalization (BN) layers to improve TTA performance. To prevent forgetting of training data, the method uses pseudo-labels derived from test samples.

**Strengths:**

1. This work addresses test-time adaptation for graph neural networks (GNNs), which is an important research direction.

2. The authors have provided code that enables reviewers to validate the proposed method.

**Weaknesses:**

1. The proposed method appears complex and lacks elegance (particularly Sec 3.1), making it difficult to understand. Providing pseudocode for the whole method might help clarify the approach and improve reader comprehension.

2. The performance improvement achieved by this method is marginal (Table 1), making it challenging to demonstrate the superiority of the approach compared to existing methods.

3. The reliance on pseudo-labels could be problematic if the pseudo-labels are inaccurate, especially in scenarios with complex distribution shifts, such as in dynamic or mixed domain shifts found in real-world or online settings like SAR. Please validate the method’s effectiveness in these challenging settings. Additionally, if optimization becomes unstable, are there any solutions to address this issue?

4. Please include an ablation study on the impact of test batch size on model performance to understand how sensitive the method is to this parameter.

**Questions:**

Please refer to the weaknesses.

---

> ### Comment · Reviewer_xPKS · 2024-11-26
>
> The authors did not provide a rebuttal, so I kept my score unchanged.

---

> ### Author Response · Authors · 2024-11-26
>
> Answer to Q1: We added pseudo-code in the Appendix A.5.
>
> Answer to Q2: Most recent methods for the TTA cannot achieve significant improvements on all datasets. Our method also does not achieve this. This is an unfortunate fact, not an excuse for us. However, we still added the ranking info of the methods in Table 1.
>
> Answer to Q3: In Section 3.2, the Entropy-based Selection and the Confidence-based Filtering are employed to select the reliable samples with pseudo-labels.
>
> Answer to Q4: We appreciate the reviewer's keen observation regarding the potential impact of test batch size on model performance. We would like to clarify that our implementation of batch normalization aligns with the established practices in GTRANS and EERM, where `BatchNorm1d` is employed as the batch normalization layer, and the entire graph is treated as a single input during training.

---

> > ### Comment · Reviewer_xPKS · 2024-11-27
> >
> > Thank you for providing the detailed rebuttal. While I appreciate the efforts made in addressing the comments, I feel that some of my concerns remain unresolved.
> >
> > Marginal Performance Improvement: The proposed method achieves only marginal performance improvement. Could you please clarify, in light of this, why the paper should be accepted at ICLR? Providing stronger evidence or additional experiments to substantiate the practical value of the method would be highly beneficial.
> >
> > Complex Distribution Shifts: The experiments provided so far do not sufficiently demonstrate the robustness of the method under complex distribution shifts, particularly in scenarios like dynamic or mixed domain shifts, which are prevalent in real-world settings (e.g., SAR). Could you provide additional results or analysis to validate the method's effectiveness in such challenging scenarios?
> >
> > Batch Size Ablation: While your response highlights the use of BatchNorm1d and its alignment with established practices, this does not fully address my concern. Could you conduct further ablation studies on the impact of test batch size on model performance? This would help elucidate the sensitivity of the method to this parameter and improve the understanding of its robustness.

---

### Official Review · Reviewer_oPAB · 2024-10-31

**Soundness:** 2
**Presentation:** 2
**Contribution:** 2
**Rating:** 3
**Confidence:** 3

**Summary:**

This work proposes a two-step batch normalization optimization method for test-time adaption in graph neural networks for better generalization performance. First, they determine weights and masks for the empirical batch mean and variance, considering training and test data statistics. Subsequently, they refine the scale and shift parameters of the BN layers using a reformulated loss function incorporating an energy-based model, aiming to enhance the model’s generalization capabilities. The experiment results show its good performance.

**Strengths:**

The method is clear, easy to understand.

**Weaknesses:**

For method: Tuning the parameter of batch normalization layer is a common idea in test-time adaption area, including optimizing parameter and statistic values. Although this paper proposes new idea of masking and model calibration, it lacks explanation/empirical results to explain the reason for their designs. It is more like an ensemble of several tricks instead of a complete algorithm. Therefore, the method is less attractive to me.

For experiment: The baseline accuracy in Table 1 is much lower than results shown in other papers[1][2], such as test accuracy on OGB-Arxiv and OGB-Products.

[1] GOAT: A Global Transformer on Large-scale Graphs
[2] POLYNORMER: POLYNOMIAL-EXPRESSIVE GRAPH TRANSFORMER IN LINEAR TIME

**Questions:**

see Weaknesses

---

> ### Author Response · Authors · 2024-11-26
>
> We appreciate the reviewer's feedback and acknowledge the relevance of Graph Transformer Networks to our research. However, we respectfully disagree with the assertion that performance is lower than GOAT and POLYNORMER.
>
> Our paper focuses specifically on TTA algorithms applied to GNNs. While GOAT and POLYNORMER utilize graph transformers, their primary contribution lies in advancing graph transformer architectures rather than addressing the challenges of TTA in GNNs.
>
> Furthermore, the experimental settings in GOAT and POLYNORMER employ random dataset splits, which may not fully capture the complexities of real-world scenarios where distribution shifts are prevalent. In contrast, our work adopts the Out-of-Distribution (OOD) setting (following the previous works in this field), which is more challenging and representative of real-world deployments.
>
> It is crucial to emphasize that a direct comparison with GOAT and POLYNORMER would be inappropriate due to the differing research focuses and experimental settings. Our work addresses the specific challenges of TTA in GNNs under OOD settings, which are not the primary focus of GOAT and POLYNORMER.

---

### Official Review · Reviewer_ikmZ · 2024-11-01

**Soundness:** 3
**Presentation:** 3
**Contribution:** 3
**Rating:** 3
**Confidence:** 3

**Summary:**

The paper proposes a novel two-step optimization method for test-time adaptation (TTA) in graph neural networks (GNNs), focusing on fine-tuning batch normalization (BN) activations. This approach effectively tackles distribution shifts, which is an important issue in GNNs, especially for practical applications involving non-stationary environments

**Strengths:**

1.	The paper proposed method is sound.
2.	Writing is good to follow.
3.	Comprehensive experiments on seven diverse datasets with different types of distribution shifts demonstrate the effectiveness of the proposed method.
4.	Large scale graphs like OGB-Products are included in the experiments.

**Weaknesses:**

1.	Only evaluate the performance on shallow GNNs. Some deep GNN should also be evaluated (e.g., GCNII).
2.	The improvement is marginal on some datasets (e.g., Twitch-E)
3.	The paper could benefit from deeper theoretical insights or a more thorough justification

**Questions:**

See the above weaknesses.

---

> ### Comment · Reviewer_ikmZ · 2024-11-27
>
> The authors did not provide a rebuttal. Therefore, my concerns have not been addressed and I decrease my score accordingly.

---

### Meta-Review · Area_Chair_tKhj · 2024-12-24

**Metareview:**

This paper proposed a two-step batch normalization optimization algorithm for  test-time adaptation for graph neural networks. This research direction is highly relevant, especially given the prevalence of distribution shifts in real-world applications.

The main concerns are on 1) marginal improvement on some datasets; and 2) novel of the paper and 3) ablation study. I hope these concerns could be addressed/clarified in the future version of this paper.

**Additional Comments On Reviewer Discussion:**

The concerns from the reviewers are not addressed during rebuttal. Authors did not actively participate in the discussion.

---

### Decision · Program_Chairs · 2025-01-22

Reject